# Removal of Total Nitrogen and Phosphorus Using Single or Combinations of Aquatic Plants

**DOI:** 10.3390/ijerph16234663

**Published:** 2019-11-22

**Authors:** Feng Su, Zhian Li, Yingwen Li, Lei Xu, Yongxing Li, Shiyu Li, Hongfeng Chen, Ping Zhuang, Faguo Wang

**Affiliations:** 1South China Botanical Garden, Chinese Academy of Sciences, Guangzhou 510650, China; sufeng15@mails.ucas.ac.cn (F.S.); lizan@scbg.ac.cn (Z.L.); liyw@scbg.ac.cn (Y.L.); xulei@scbg.ac.cn (L.X.); liyongxing.7@163.com (Y.L.); lishiyu@scbg.ac.cn (S.L.); h.f.chen@scbg.ac.cn (H.C.); 2University of Chinese Academy of Sciences, Beijing 100049, China; 3Southern Marine Science and Engineering Guangdong Laboratory (Guangzhou), Guangzhou 511458, China; 4Key Laboratory of Plant Resources Conservation and Sustainable Utilization, Guangdong Provincial Key Laboratory of Applied Botany, South China Botanical Garden, Chinese Academy of Sciences, Guangzhou 510650, China

**Keywords:** aquatic plant, eutrophication, nitrogen, phosphate, removal, wastewater

## Abstract

Phytoremediation is a potentially suitable technology for taking up large amounts of N and P during plant growth and the removal of plant material, thereby avoiding eutrophication. We compared the capacity of nine different aquatic plant species for removing total P (TP), total N (TN), and NH_4_^+^-N from raw domestic sewage wastewater collected from a living area located in Guangzhou city, China, and different concentrations of artificial wastewater. The experiments were performed in two stages, namely screening and modification. In the screening stage, four plant species were identified from the nine grown in raw domestic sewage water for 36 days. In the modification stage, the TN and TP removal ability of different plant combinations were determined in artificial wastewater at different N/P concentrations. After having been grown in monocultures for 46 days, *Ipomoea aquatica* (90.6% and 8.8%) and *Salvinia natans* (67.3% and 14.2%) obtained the highest TP removal efficiency in lightly and highly polluted wastewater, respectively. The combination of *S. natans* and *Eleocharis plantagineiformis* effectively removed TP and TN from lightly polluted water, suggesting that this combination is suitable for phytoremediation of eutrophic wastewater.

## 1. Introduction

With the rapid development of the Chinese economy and the accompanying accelerated urbanisation, various pollutants have been discharged into surface water bodies, which has led to increasingly serious pollution of rivers and lakes [1]. According to national surveys, the water area currently affected by persistent pollution and eutrophication in China is vast [2]. Eutrophication endangers aquatic organisms and leads to the degradation of aquatic ecosystems, a decline in biodiversity, the collapse of nutrient cycles, and the development of water blooms, which negatively affect agriculture, industry, and drinking water production [3,4]. Eutrophication is mostly the result of human activities that release large amounts of N and P into water bodies [3]. Pollutants enter water bodies through surface runoff, erosion, rainfall, and groundwater [5]. For example, animal production, agricultural practices, and endogenous pollution can result in an extensive release of N and P, considerably influencing the water environment [2,6].

To address these serious issues, various measures to remove pollutants have been developed for the eco-remediation of eutrophic water bodies, not only in China but also abroad [1,4,7]. Current remediation technologies include physical, chemical, and engineering methods, but they are either too expensive or labour-intensive. For example, chemical remediation has the advantages of low energy consumption, low cost and quick effect; however, it has the disadvantages that chemical agents may increase the toxicity of water body, and chemical agents may cause secondary pollution of the water body [8]. Aquatic plant restoration (phytoremediation) has been receiving increasing attention, as water plants can take up and store large amounts of N and P [9,10]. Different studies on nutrient removal using phytoremediation with aquatic plants have been performed with respect to the efficiency of restoration of sewage wastewater [11,12,13], aquaculture pond effluent, dairy effluent, and livestock wastewater [9,14] as well as the development of synthetic nutrient solutions [15]. Different aquatic plant species, such as water hyacinth (*Eichhornia crassipes*), duck weed (*Lemna* spp.), water lettuce (*Pistia stratiotes*), vetiver grass (*Chrysopogon zizanioides*), and common reed (*Phragmites australis*), have been widely used to remove nutrients from polluted river water or wastewater [11,16,17,18]. For instance, Lu et al. [11] reported the great potential of *P. stratiotes* to remove N and P from various types of polluted water. Moreover, total N (TN, 63.2%) and total P (TP, 36.2%) were removed by remediating treated swine wastewater with *Pistia stratiotes* and *Lemna* sp., respectively [19]. Iamchaturapatr et al. [20] tested 21 plant species and suggested that the N and P removal capacity depended on the species. Most previous studies reported on the remediation efficiency of polluted rivers or wastewater using single plant species. However, the removal mechanisms of water nutrients using combinations of different aquatic plant species and the subsequent remediation of polluted water need to be more thoroughly explored [21].

The aim of this study was, therefore, to examine the ability of nine aquatic plants to remove TN and TP from wastewater at the screening and modification stages. In this study, a screening experiment was conducted in which plant species were grown in raw sewage water over a period of 36 days while the N and P contents in the water were monitored. A modification experiment was also conducted, wherein various combinations of plant species were tested for their ability to purify artificial domestic sewage or animal farm wastewater over a period of 46 days.

## 2. Materials and Methods 

### 2.1. Plant Species

The plant species used in this study were *Salvinia natans*, *Eleocharis plantagineiformis*, *Ipomoea aquatica*, *Hydrocotyle vulgaris*, *Colocasia tonoimo*, *Typha orientalis*, *Eichhornia crassipes*, *Dysophylla sampsonii*, and *Rotala indica*, which were selected from more than 30 aquatic plant species based on their good growth rates. Namely, in this preliminary study, we tested over 30 aquatic plants collected from the field sewage ditch in the Pearl River Delta. After a period of indoor adaptive cultivation, we found that some aquatic plants did not grow well or grew very slowly, and then kept the fast-growing species for further research. These nine species were used in the first stage of the experiments (Table 1). They were introduced from a polluted riverside in the Pearl River Delta in Guangdong Province. *Hydrocotyle vulgaris*, a common invasive species that is already widely distributed in southern China, was included as a control. The plants were cultivated in a 30 L container and placed in a greenhouse for two weeks. All the selected plants were cultured in a ¼-strength modified Hoagland’s nutrient solution [22]. Water was added every three days to maintain the content. Hoagland’s nutrient solution was added once a week to maintain adequate nutrient content for plant growth.

### 2.2. Wastewater

The raw sewage water used in the first stage of the experiments was collected from a drainage ditch (Cencun River) flowing through the South China Botanical Garden, Guangzhou. The source of this water body is mainly domestic sewage directly discharged by nearby residents (Appendix A). The concentrations of relevant pollutants in and the pH of the sewage water are listed in Table 2. The second stage of the experiments was performed with artificial eutrophic water, resembling either domestic sewage water or animal farm wastewater. The ingredients and final concentration of pollutants are summarised in Table 2. Small amounts of Ca and Mg were added to support plant growth.

### 2.3. Experimental Design

All experiments were conducted in a greenhouse located in the South China Botanical Garden. The flowchart of this study is presented in Appendix A. The indoor temperature was not regulated, and natural light was available through the glass. For the first-stage experiments, 6–10 individual plants of the same species with similar individual biomass were selected. They were weighed to record the initial overall wet biomass, after which they were transferred to a hydroponic basin containing 12 L untreated sewage water. The experiment with each plant species was performed in triplicate (Figure 1). During these experiments, water samples were collected on days 1, 6, 11, 16, and 36. At the end of the experiments (day 36), the plants were weighed to measure the wet biomass.

The second-stage experiments were restricted to four species (Table 1), of which the plants were tested singly and in the following combinations: *S. natans–E. plantagineiformis* (number 14), *H. vulgaris–E. plantagineiformis* (number 15), *I. aquatica–E. plantagineiformis* (number 16), and *S. natans–E. plantagineiformis–H. vulgaris* (number 17). In both sets of experiments, a control without plants was included. Six to ten uniform plants were transferred to larger hydroponic tanks containing 30 L of artificial sewage. Water was sampled on days 1, 6, 11, 16, 36, and 46, and the leaves of single plants were harvested on days 11 and 36, after which the plants continued to grow until day 46.

### 2.4. Sample Preparation and Analytical Procedures

The roots and leaves of the tested plants were separated, washed three times with deionised water, and dried to a constant weight at 65 °C [7]. The dried plant material was ground, and the P and N contents were measured. The studied water quality indicators were pH and TN, TP, and NH_4_^+^-N concentrations. The TN concentration in the water was determined using KCL extraction–indophenol blue colourimetry with a TOC analyser (TOC-VCSH, Shimadzu) according to the national guideline GB 11894-89. TP was determined using molybdenum antimony spectrophotometry, as per guideline GB 11893-89, and NH_4_^+^-N was determined using the salicylic acid–hypochlorite spectrophotometric method, as per GB 7481-87. The total amount of each pollutant removed from the water was calculated as the difference between the initial and final concentrations in the water. Pollutant removal rates were expressed as a percentage relative to the initial values.

At the beginning of the first experiments, shoots were cut from the plants for drying and grinding, and the TP and TN content per gram of the dried material was determined. At the end of the experiments, these analyses were done for both shoots and roots. The performance of the plants in the second experiments was measured by comparing their TN and TP contents at the beginning and end of the experiments, and the pollutant removal was expressed as a percentage (%) relative to the initial pollutant concentrations in the water. All experiments were performed in triplicate, and the average results per data point are presented.

### 2.5. Statistical Analyses

All data were statistically analysed via one-way analyses of variance using SPSS 19.0 software (SPSS Inc., Chicago, IL, USA), and significant differences were tested using the least significant difference and Duncan multiple comparisons. Standard errors obtained with the triplicate experiments were graphically shown when they exceeded 5%.

## 3. Results

### 3.1. Comparison of Single Plant Species

During the growth of the various single plant species for 36 days, the four water quality parameters were measured at five time points. The pH of the control water without plants increased over time, from 7.86 at the beginning of the experiment to 8.80 on day 36 (Figure 2A). *Typha orientalis* caused the lowest pH value (7.28), and the presence of *H. vulgaris* also significantly lowered the pH. The other plant species caused fluctuations in pH, with an overall increase during the first 16 days followed by a slight decrease towards day 36.

During the first six days of the experiment, the NH_4_^+^-N concentration in the water rapidly decreased in all cases, even in the control (although the minimum was reached after 16 days) (Figure 2B). The TN content also decreased rapidly (Figure 2C), with slight variations between days 6 and 16, depending on the plant species. All TN levels remained consistently low after day 16 at approximately 10%–20% of the original levels; this was also observed in the plant-free control. 

The TP content decreased during the first six days, but the extent of this decrease depended on the presence of plants, and large differences between the plant species were observed. Notably, *D. sampsonii* and *R. indica* did not perform as well as the other plants did. A temporary peak at day 11 was observed in all experiments, after which the TP levels decreased again. In the presence of *T. orientalis*, the TP levels significantly increased between days 16 and 36. A weaker upward trend during this period was also seen for most other plants, with the exception of *R. indica*, *D. sampsonii*, and *I. aquatica* (Figure 2).

The changes in the wet biomass of the aquatic plants during the 36 days of treatment are summarised in Appendix A. To some extent, biomass production depends on the ability of plants to adapt to the culturing conditions. *Hydrocotyle vulgaris* and *S. natans* grew well, resulting in the highest absolute and relative increases in wet biomass. Not all plants obtained net growth, and four species decreased in biomass. The growth of *E. crassipes*, which can be aggressive, was the highest of all species, but it was not exceptional, since its growth may have been limited by spatial constraints. 

The TN and TP contents of the plant material were determined by analysing plant shoots at the beginning of the experiments and those harvested at day 36 (Table 3). Generally, the N content was higher than that of P. The initial TN content was highest for *I. aquatica* and lowest for *E. plantagineiformis* (*p* < 0.05). For most species, TN content did not differ significantly between the stems/leaves and roots at the end of the experiments, with the exception of *D. sampsonii*, which had higher TN levels in the roots, and *E. plantagineiformis*, which had higher TN levels in the stems and leaves. On day 36, the N content of the stems and leaves of *C. tonoimo* was the highest, and that of *H. vulgaris* was lowest. The initial TP content was significantly higher for *C. tonoimo* than for the other plant species (*p* < 0.05), whereas *D. sampsonii* had the lowest initial P content. *Salvinia natans* had the highest P content in the stems and leaves, whereas *R. indica* had the lowest values. For roots, the maximum and minimum P contents were found in *E. crassipes* and *H. vulgaris*, respectively (there was not enough root material to analyse that of *S. natans*).

### 3.2. Combinations of Plant Species

In the second-stage experiments, four plant species selected from the first-stage experiments were tested singly and in combination for the phytoremediation of artificial urban sewage and animal farm wastewater, in which the pollutant concentration is typically high. The two types of wastewater were produced, as described in Section 2.2, and the experiment was extended for 46 days.

The pH of both types of wastewater decreased more drastically, to below 5, than that in the first-stage experiments (Figure 3A). The variation in pH between triplicate experiments was extensive in many cases, and the control and plant treatments did not differ significantly. This poor reproducibility might be due to the lack of buffering capacity in the artificial water. In the last 10 days of the experiments, the pH did not vary significantly within treatments, suggesting that an equilibrium had been reached (Figure 3B).

In the presence of a single plant species, the NH_4_^+^-N concentration decreased to 6.6–8.5 mg/L on day 46, which was significantly lower than that of the control (Figure 3C). The combinations of two plant species (numbers 14 and 16) caused the lowest NH_4_^+^-N concentrations, with the *H. vulgaris*–*E. plantagineiformis* combination performing best. The lowest overall NH_4_^+^-N level was obtained with the *I. aquatica*–*E. plantagineiformis* combination. Notably, the artificial wastewater experiments allowed a more accurate assessment of the contribution of the plants (as well as their rhizobiomes) in NH_4_^+^-N removal.

The TN content of the artificial domestic water was removed over time in a plant-dependent manner, with the *H. vulgaris*–*E. plantagineiformis* combination (No. 15) performing best (Figure 4A). However, the high TN concentration in the artificial animal farm wastewater could not be effectively removed (Figure 4B), indicating that a considerable fraction of the decreased NH_4_^+^-N had been converted to other forms, possibly nitrates.

The TP content of the artificial domestic wastewater decreased in all tests to reach lower values than those of the control after 46 days (Figure 4C). The high phosphate concentration of the artificial animal farm wastewater could be decreased to a level as low as 17.3 mg/L. Combination treatments Nos. 14, 16, and 17 produced significantly lower TP concentrations than the control did (Figure 4D). 

The biomass of the plants was determined at three time points: days 11, 36, and 46. For the initial two time points, individual leaves were cut, weighed, and analysed, after which the plants continued to grow. These weights were added to that of the plants harvested on day 46, and the changes in the overall biomass of the plants were calculated (Appendix A). During the first few weeks, the plants had to adapt to the new growth conditions, resulting in large differences between the two artificial wastewater types, but this difference became smaller by day 46. Of the four monocultures, *S. natans* produced the strongest increase in biomass, but this increase was halved in the animal farm wastewater compared to that in artificial domestic wastewater. 

The removal rate was calculated for the entire experimental period of 46 days and expressed as a percentage of the initial pollutant concentration (Table 4). For TP, high removal rates (67.3%–90.6%) were obtained in the artificial domestic wastewater with all plant species, except for *E. plantagineiformis*. The highest removal rate obtained with an *I. aquatica* monoculture. From the phosphate-rich artificial animal farm wastewater, up to 14% of P could be removed by an *S. natans* monoculture, and the triple combination resulted in 33.7% TN removal. The highest TN removal rate in artificial domestic wastewater (20.3%) was obtained with *E. plantagineiformis*, whereas this species could only remove 5.9% of TN from artificial animal farm wastewater as a monoculture or in combination with *S. natans*. The three other plant species or combinations could remove around 15% of TN from artificial domestic wastewater (*S. natans* alone and the two combinations containing *E. plantagineiformis*), but none of the tests resulted in more than 5.9% TN removal from the artificial animal farm wastewater (Table 4).

## 4. Discussion

The results of this study showed that the nine aquatic plant species or their combinations played a notable role in water purification within 36 or 46 days. In general, the biomass of the studied plants increased in different pollutant treatments, indicating that they had strong ecological adaptability to N and P nutrient stress. In the screening stage with raw sewage water, the plants caused an increase in the pH, whereas they tended to cause a decrease in the pH of the two types of artificial domestic wastewater, suggesting that different aquatic plants have different responses to pH due to their different physiological characteristics. Hu et al. [23] compared the purification abilities of five plants (*Azolla imbricata*, *S. natans*, *Marsilea quadrifolia*, *Hydrilla verticillate*, *and*
*Sagittaria sagittifolia*) in lightly, moderately, and highly eutrophic water and found that, under various eutrophic conditions, all plants reduced the pH of water, except for the *Hydrilla* system, which was in line with our results. The screening experiment showed that the most suitable species for removing N and P were *E. plantagineiformis*, *H. vulgaris*, *I. aquatica*, and *S. natans*, or their combinations. With regard to emergent plants (e.g., *E. plantagineiformis*), they have strong roots, which can effectively absorb nutrients in the wastewater [24]. While the floating plants (e.g., *S. natans*) have developed roots, which can effectively absorb nutrients in the water body [24]. The TN and TP removal rates of the nine plant species in eutrophic water were higher during the first six days. The main reason was that the roots of the newly transplanted plants were damaged, and new root growth was promoted. Subsequently, the plants adapted to the environment and exhibited normal growth, after which the TN and TP absorption tended to normalise. 

Because aquatic plants absorb N, P, and other elements during their growth process, nutrients can be removed from wastewater through harvesting to consequently purify eutrophic water [11]. Several studies have shown effective methods of N removal in wastewater, including assimilation and absorption by plants, NH_4_^+^-N volatilisation, and nitrification and denitrification [25]. The different plants differed in their effect on TN removal. This difference was more obvious in the treated sewage water, but the NH_4_^+^-N in the *E. plantagineiformis*–*H. vulgaris* combination was higher than that in the control. Cui et al. [26] reported that the removal rates of NH_4_^+^-N in Phnom Penh *Chlorophytum*, *Herba Lysimachiae*, and *Scindapsus aureus* were 46%, 60%, and 70%, respectively. In the present study, four single plants or their combinations achieved only 0.6%–33% TN removal in the two artificial sewage water environments, suggesting that plant uptake has little effect on TN removal. It is worth noting that the TN content in the control decreased over time and stabilised on day 16, which indicated that the synergistic effect of microorganisms in the water resulted in self-purification. The volatilisation of NH_4_^+^-N can only occur under conditions with pH >8. Plants can directly absorb soluble N and P and provide a medium for microbial growth [19]. Plant root exudates can further promote microorganism growth and increase nitrification and denitrification rates, thereby improving the purification rate of eutrophic water [27]. For the simulated wastewater with high N and P concentration, the decrease rate of TN and TP is small when compared to the control (Figure 4). This may be because the plant absorption capacity of pollutants is limited, which leads to a low removal rate in a short treatment time; moreover, it may also due to the high concentration of N and P has caused stress on plants, thus affecting the purification efficiency of plants. From the results of this study, the N assimilation of *I. aquatica* in the moderately and highly eutrophic water was the highest, but the assimilated N accounted for only 4.39%–7.27% of the TN removal. Therefore, the plants achieved N purification by promoting nitrification and denitrification, which may be the main pathway of N removal. This may also be related to the difference in the forms of N utilisation, the utilisation efficiency of different plants, and the total biomass, which deserves further research and analysis. 

In this study, the TP removal rate was 21%–91% with all studied plant species in artificial domestic wastewater, which was similar to the findings by Lu et al. [11], who reported that TP removal through absorption by *E. crassipes*, *P. stratiotes*, and *Myriophyllum spicatum* was 58%, 64%, and 83%, respectively. Torit et al. [28] showed that the TP removal by *Echinodorus cordifolius* from domestic wastewater was 16%. Similar results were found in the present study, with a TP removal of 6.7%–34% of the studied plants. Figure 4 shows that the contribution rate of plants to TP removal in different concentrations of wastewater differed, indicating that the concentration of pollutants in water has a negative effect on the response of plants. However, for the control (with no plants) in this study, the decrease in TP is likely to be the result of microbial synergies. Generally, the TN and TP removal rate in low-concentration wastewater was much higher than that in high-concentration wastewater. The TP removal efficiency in water was higher than that of TN, and the peak period of purification efficiency occurred during the first 36 days. The TN and TP removal rates are compared in Table 4, showing that plants contributed more to TP removal than to TN removal in water. Therefore, in the process of TP removal, plants with high TP uptake should be selected for phytoremediation.

Our results are similar to those of previous studies conducted with other plant species [11,29], but the species we selected had the advantage of removing TP. We found that combinations of plant species did not perform better than monocultures did. Compared with traditional water body remediation technology, phytoremediation is relatively cost-effective and less labour-intensive in terms of operation and management, and it can, therefore, be profitable [30]. One limitation of the approach is that it needs more time for effective water treatment [31] and that the effectiveness may fluctuate seasonally and with plants that may not cope well with highly polluted water. In addition, growing plants must be harvested in a timely manner to avoid their death and decomposition, which would return the stored nutrients to the water.

In this study, artificial simulation of eutrophication water body was used in an indoor environment, which was different from the physical and chemical properties of the actual polluted water body, ignoring the release of pollutants in sediment and the impact of weather or external environment. Additionally, the aquatic plants were harvested more frequently in this experiment, resulting in an overall yield not as good as expected. Thus, in the follow-up study, the optimal restoration species or combination could be considered to be applied to the actual eutrophication water, and the in-situ ecological restoration of eutrophication water should be investigated.

## 5. Conclusions

Plants can play a significant role in purifying eutrophic water. The main TN and TP removal pathways involve the synergistic effects of plant roots and microorganisms, some of which are removed via microbial activity, whereas only a few are absorbed directly by plants. The results of this study showed that plants contribute more to TP removal than to TN removal. Therefore, plants with high TP uptake have great application value in phytoremediation. In terms of nutrient accumulation, *I. aquatica* and *E. plantagineiformis* had the best purification effect for TP and TN, respectively. Water bodies containing high concentrations of these pollutants require treatment with a combination of *S. natans*, *E. plantagineiformis*, and *H. vulgaris*. In conclusion, our results indicated that various aquatic plant species have great potential in the purification of eutrophic water.

## Figures and Tables

**Figure 1 ijerph-16-04663-f001:**
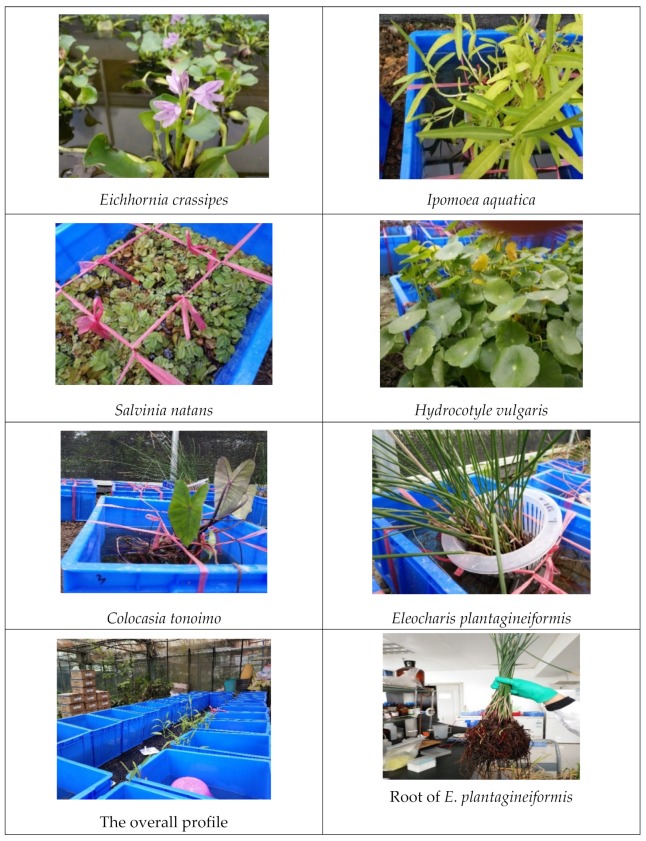
The phenomenon of selected aquatic plants.

**Figure 2 ijerph-16-04663-f002:**
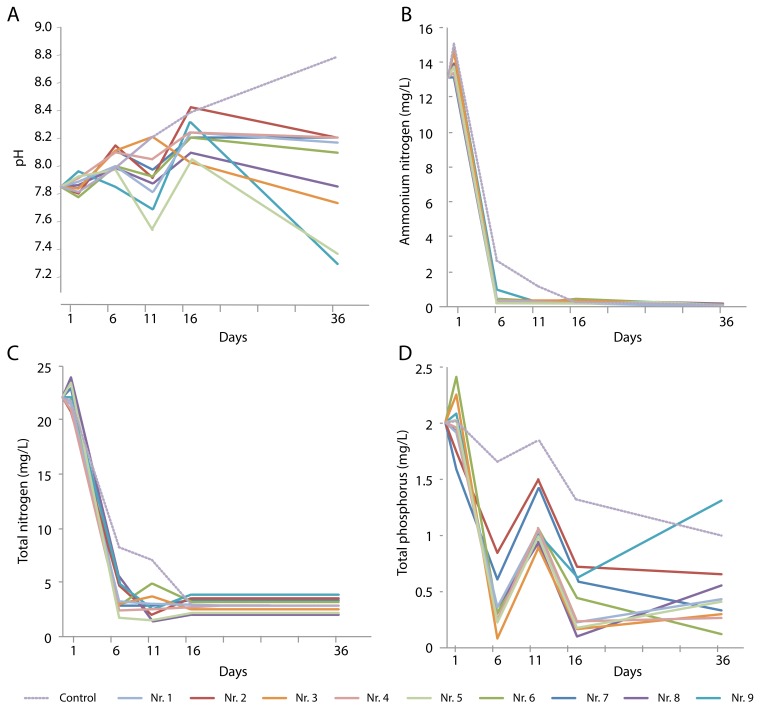
Changes over time in (**A**) pH and content of (**B**) NH_4_^+^-N, (**C**) total N, and (**D**) total P in sewage water in the presence of various plant species: No. 1, *Colocasia tonoimo*; No. 2, *Dysophylla sampsonii*; No. 3, *Eichhornia crassipes*; No. 4, *Eleocharis plantagineiformis*; No. 5, *Hydrocotyle vulgaris*; No. 6, *Ipomoea aquatica*; No. 7, *Rotala indica*; No. 8, *Salvinia natans*; No. 9 *Typha orientalis*.

**Figure 3 ijerph-16-04663-f003:**
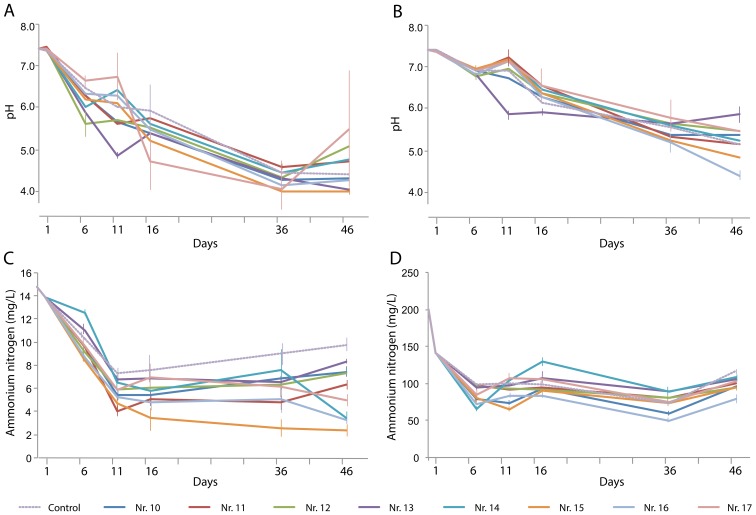
Changes over time in (**A**,**B**) pH and (**C**,**D**) NH_4_^+^-N content in (**A**,**C**) artificial domestic sewage water and (B, D) artificial animal farm wastewater in the presence of various plant species and their combinations: No. 10, *Eleocharis plantagineiformis*; No. 11, *Hydrocotyle vulgaris*; No. 12, *Ipomoea aquatica*; No. 13, *Salvinia natans*; No. 14, *S. natans–E. plantagineiformis*; No. 15, *H. vulgaris–E. plantagineiformis*; No. 16, *I. aquatica–E. plantagineiformis*; No. 17, *S. natans–E. plantagineiformis–H. vulgaris*. Mean results of three replicate experiments are shown, with standard errors of >5%. For clarity, standard errors of pH results are only shown for maximum and minimum values per time point.

**Figure 4 ijerph-16-04663-f004:**
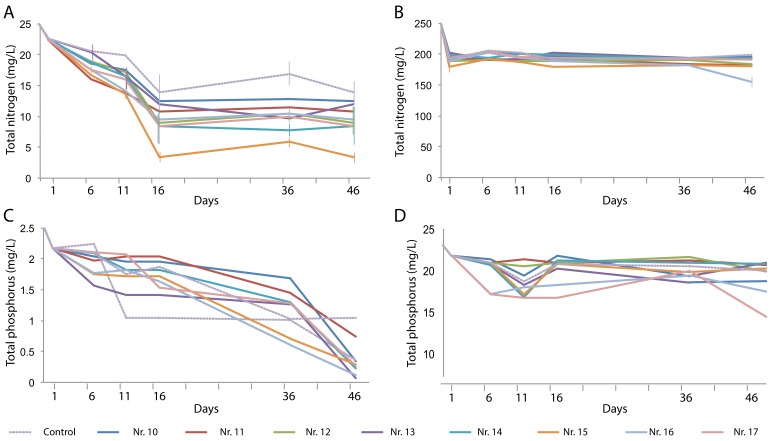
Changes over time in total N (**A**,**B**) and total P (**C**,**D**) content in artificial domestic sewage water (**A**,**C**) and artificial animal farm wastewater (**B**,**D**) in the presence of various plant species and their combinations. Plant species are the same as those in Figure 3.

**Table 1 ijerph-16-04663-t001:** Aquatic plant species used in this study.

Scientific Name	Common Name	Family	Experiment Number	
			Stage 1	Stage 2
*Colocasia tonoimo*		Araceae	1	
*Dysophylla sampsonii*		Lamiaceae	2	
*Eichhornia crassipes*	Water hyacinth	Pontederiaceae	3	
*Eleocharis plantagineiformis*		Cyperaceae	4	10, 14, 15, 16, 17
*Hydrocotyle vulgaris*	Water pennywort	Apiaceae	5	11, 15
*Ipomoea aquatica*	Swamp morning glory	Convolvulaceae	6	12, 16
*Rotala indica*	Indian toothcup	Lythraceae	7	
*Salvinia natans*		Salviniaceae	8	13, 14, 17
*Typha orientalis*	Bull rush	Typhaceae	9	

**Table 2 ijerph-16-04663-t002:** Basic physical and chemical properties of the domestic wastewater and the ingredients in artificial wastewater.

Concentration (mg/L)	NH_4_^+^-N	TN		PO_4_^3–^P	TP		pH
Domestic sewage	13.16	22.1		N.D.	2.0		7.86
Artificial domestic sewage	15.0	25.0		2.0	2.5		7.4
Artificial animal farm wastewater	200	250		15	23		7.4
National standard for Class V water ^a^	2.0	2.0			0.2–0.4		6-9
**Ingredients (g/L)**	**Ca(NO_3_)_2_·4H_2_O**	**K_2_NO_3_**	**NH_4_Cl**	**KH_2_PO_4_**	**Urea**	**CaCl_2_**	**MgCl_2_**
Artificial domestic wastewater	62.4	72.6	105	21.9	16.2	0.004	0.004
Artificial animal farm wastewater	627	728	1055	234	162	0.022	0.022

Note: ^a^ environmental quality standard for surface water established by China (GB3838-2002).

**Table 3 ijerph-16-04663-t003:** Total nitrogen (TN) and total phosphorus (TP) content in plant shoots and roots (mg g^−1^ dry weight, mean ± standard error, *n* = 3).

Plants	TN			TP		
	Initial Content (Shoots)	Content at Day 36		Initial Content	Content at Day 36	
		Shoots	Roots		Shoots	Roots
*Colocasia tonoimo*	3.43 ± 0.24 ^d^	4.40 ± 0.54 ^a^	4.28 ± 0.40 ^a^	4.69 ± 0.24 ^a^	2.15 ± 0.13 ^abc^	1.40 ± 0.14 ^ab^
*Dysophylla sampsonii*	2.59 ± 0.30 ^e^	2.81 ± 0.06 ^bc^	3.66 ± 0.12 ^a^	0.19 ± 0.01 ^f^	0.55 ± 0.08 ^bc^	0.84 ± 0.10 ^ab^
*Eichhornia crassipes*	4.28 ± 0.09 ^bc^	4.24 ± 0.13 ^a^	3.70 ± 0.21 ^a^	0.17 ± 0.01 ^f^	2.72 ± 0.04 ^b^	1.79 ± 0.26 ^a^
*Eleocharis plantagineiformis*	1.62 ± 0.11 ^f^	3.28 ± 0.10 ^b^	2.31 ± 0.09 ^b^	0.26 ± 0.03 ^ef^	1.03 ± 0.03 ^ab^	0.70 ± 0.14 ^b^
*Hydrocotyle vulgaris*	3.61 ± 0.11 ^cd^	1.98 ± 0.18 ^d^	1.76 ± 0.13 ^b^	1.80 ± 0.19 ^c^	1.45 ± 0.12 ^a^	0.22 ± 0.01 ^ab^
*Ipomoea aquatica*	5.10 ± 0.53 ^a^	4.11 ± 0.00 ^a^	4.11 ± 0.00 ^a^	0.64 ± 0.08 ^def^	0.80 ± 0.45 ^abc^	1.33 ± 0.77 ^ab^
*Rotala indica*	3.61 ± 0.20 ^cd^	3.18 ± 0.33 ^b^	3.70 ± 0.64 ^a^	1.03 ± 0.05 ^d^	0.21 ± 0.08 ^c^	0.99 ± 0.19 ^ab^
*Salvinia natans*	4.47 ± 0.14 ^ab^	4.26 ± 0.37 ^a^	-	2.65 ± 0.34 ^b^	3.24 ± 0.50 ^b^	-
*Typha orientalis*	3.33 ± 0.18 ^d^	2.38 ± 0.01 ^cd^	2.49 ± 0.05 ^b^	0.77 ± 0.15 ^de^	0.81 ± 0.04 ^abc^	0.54 ± 0.08 ^ab^

Values among the plant species having the same letter are not significantly different (*p* = 0.05).

**Table 4 ijerph-16-04663-t004:** Removal rate of pollutants by different plant species after 46 days of treatment (mean ± standard error, n = 3).

Plants	Plant Removal Rate of Total Phosphorus		Plant Removal Rate of Total Nitrogen	
	Artificial Domestic Wastewater	Artificial Animal Farm Wastewater	Artificial Domestic Wastewater	Artificial Animal Farm Wastewater
*Eleocharis plantagineiformis*	21.2% ^e^	6.7% ^ab^	20.3% ^a^	5.0% ^ab^
*Hydrocotyle vulgaris*	84.4% ^ab^	8.3% ^ab^	6.9% ^b^	-
*Ipomoea aquatica*	90.6% ^a^	8.8% ^bc^	8.7% ^b^	0.6% ^de^
*Salvinia natans*	67.3% ^d^	14.2% ^bc^	14.1% ^b^	2.2% ^c^
*S. natans–E. plantagineiformis*	79.0% ^bc^	7.3% ^a^	15.7% ^b^	5.9% ^a^
*H. vulgaris–E. plantagineiformis*	82.8% ^abc^	6.7% ^ab^	13.6% ^b^	0.9% ^d^
*I. aquatica–E. plantagineiformis*	74.1% ^cd^	20.1% ^bc^	7.0% ^c^	1.6% ^cd^
*Salvinia natans–E. plantagineiformis–H. vulgaris*	88.4% ^ab^	33.7% ^a^	5.9% ^c^	4.0% ^b^

Values among the plant species having the same letter are not significantly different (*p* = 0.05).

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
