# Peer review of "Removal of Total Nitrogen and Phosphorus Using Single or Combinations of Aquatic Plants"

_ijerph, 2019, doi:10.3390/ijerph16234663_

Round 1

Reviewer 1 Report

The use of plants to treat waste water is not really a novelty !

The interest here is to compare single species to mixed ones.

But the work is not enough detailed about what it brings as new data.

Need some more discussion and explanations about some physiological interpretations, if not through their own measures, at least through (existing) bibliographical data.

How can You explain the P peak after 11 days in exp 1

Where is P ? lower in plants and in waste water ?

Explications ?

Reviewer 2 Report

Dear authors,

I read this interesting paper about plan remediation techniques. I consider that it is very well-written and structured. However, I have some concerns related to the explanation of the methods and figures. I suggest including a flowchart in order to make more visible your paper. Also, for me, it would be mandatory to include photos of the experiments, lab work and plots. I included some tips to improve your figures. Please, see the attached pdf. I also suggest including a paragraph in the discussion related to the challenges observed during the monitoring period and for the future. Moreover, it would be nice if you include more details about the study area, data on water pollution, etc. Now, there is a poor context.

Round 2

Reviewer 1 Report

OK for answering the questions and suggestions.

Reviewer 2 Report

Dear authors,

I am happy to see that you improved a lot of your paper. Please, consider adding the 3 figures in the main text, not as suppl. material. They are very nice!

Author Response

Thank you for your valuable suggestion. We have changed figure S3 into figure 1 in the main text.